# Tegument Protein pp150 Sequence-Specific Peptide Blocks Cytomegalovirus Infection

**DOI:** 10.3390/v13112277

**Published:** 2021-11-15

**Authors:** Dipanwita Mitra, Mohammad H. Hasan, John T. Bates, Gene L. Bidwell, Ritesh Tandon

**Affiliations:** 1Department of Microbiology and Immunology, University of Mississippi Medical Center, 2500 North State Street, Jackson, MS 39216, USA; dmitra@umc.edu (D.M.); mohammad_hasan@brown.edu (M.H.H.); jtbates@umc.edu (J.T.B.); 2Department of Medicine, University of Mississippi Medical Center, 2500 North State Street, Jackson, MS 39216, USA; 3Department of Neurology, School of Medicine, University of Mississippi Medical Center, Jackson, MS 39216, USA; gbidwell@umc.edu; 4Department of Cell and Molecular Biology, University of Mississippi Medical Center, Jackson, MS 39216, USA; 5Department of Pharmacology and Toxicology, University of Mississippi Medical Center, Jackson, MS 39216, USA; 6Biomolecular Sciences, School of Pharmacy, University of Mississippi, Oxford, MS 38677, USA

**Keywords:** CMV, herpesviruses, peptide therapy, tegument, nuclear egress

## Abstract

Human cytomegalovirus (HCMV) tegument protein pp150 is essential for the completion of the final steps in virion maturation. Earlier studies indicated that three pp150nt (N-terminal one-third of pp150) conformers cluster on each triplex (Tri1, Tri2A and Tri2B), and extend towards small capsid proteins atop nearby major capsid proteins, forming a net-like layer of tegument densities that enmesh and stabilize HCMV capsids. Based on this atomic detail, we designed several peptides targeting pp150nt. Our data show significant reduction in virus growth upon treatment with one of these peptides (pep-CR2) with an IC_50_ of 1.33 μM and no significant impact on cell viability. Based on 3D modeling, pep-CR2 specifically interferes with the pp150–capsid binding interface. Cells pre-treated with pep-CR2 and infected with HCMV sequester pp150 in the nucleus, indicating a mechanistic disruption of pp150 loading onto capsids and subsequent nuclear egress. Furthermore, pep-CR2 effectively inhibits mouse cytomegalovirus (MCMV) infection in cell culture, paving the way for future animal testing. Combined, these results indicate that CR2 of pp150 is amenable to targeting by a peptide inhibitor, and can be developed into an effective antiviral.

## 1. Introduction

Human Cytomegalovirus (HCMV), a betaherpesvirus, infects the majority of the world’s population but acute disease is manifested only in a small proportion of infected individuals [1]. Primary infection or reactivation of latent virus can cause life-threatening complications in immunocompromised individuals such as AIDS patients and transplant recipients. HCMV is also the leading cause of congenital infections and is associated with a multitude of cardiovascular diseases [2,3]. Currently, there are no commercially available vaccines for HCMV infection. Antivirals are available; however, adverse side effects and drug resistance are a growing concern.

HCMV has a prototypical herpesvirus virion (infectious virus particle) that consists of an external membranous envelope, a proteinaceous layer called the tegument and an icosahedral capsid that contains a compressed, large (>240 kb) double-stranded DNA genome. Like all herpesviruses, HCMV maturation occurs in two distinct phases, primary and secondary maturation [4]. Primary maturation begins in the host-cell nucleus where viral genome replication, capsid assembly and encapsidation take place [5]. After assembly, the nucleocapsids (NC) migrate from the nucleus to the cytoplasm. During this nuclear egress, the nucleocapsid first undergoes a primary envelopment at the inner nuclear membrane, then traverses through the nuclear envelope, followed by de-envelopment at the outer nuclear membrane, and finally reaches the cytoplasm where it accumulates within a ring-shaped perinuclear structure known as the cytoplasmic virus assembly compartment (vAC) where the secondary or final steps of virion maturation occur [4,5,6,7,8,9].

The virion maturation process is characterized by numerous virus–virus, host–host and host–virus interactions, and the tegument proteins play critical roles during this process [1,8,9]. The tegument proteins are also active during viral entry as they are released into the cells along with viral capsids and aid in viral gene expression, virus replication and evasion of host immune responses [1,10]. During late stages of infection, tegument proteins accumulate to high levels in the vAC and contribute to virion maturation and egress [8,9,10]. Thus, tegument proteins offer important targets for the development of antiviral therapy against HCMV infection.

The HCMV UL32 gene encodes a prominent betaherpesvirus-conserved virion tegument protein, pp150 (basic phosphoprotein/ppUL32), which is initially expressed in the nucleus but accumulates within the vAC during late stages and supports the final steps in virion maturation [11,12]. To date, several studies have been conducted to understand the structure and function of pp150 [11,12,13,14,15,16,17,18,19,20,21]. It is well-established that pp150 controls cytoplasmic events during virion maturation [12,14,15]. The principal role of pp150 is to stabilize and retain the nucleocapsid organization throughout the final or secondary envelopment inside the vAC [8,11,12]. The cryoEM reconstructions of HCMV virion show that pp150 is arranged in upper and lower helix bundles, which are joined by a central helix [16,18,22,23,24]. The HCMV capsid is an ensemble of 60 asymmetric units, with each of these units containing (i) 16 copies of the major capsid protein (MCP), which exists in penton and hexon capsomers, (ii) 16 copies of small capsid proteins (SCP) that sit atop each MCP, (iii) Five and one-third heterotrimeric triplexes (Ta, Tb, Tc, Td, Te and Tf, respectively) composed of the triplex monomer protein Tri1 (minor capsid-binding protein), which is coupled with triplex proteins Tri2A and Tri2B (also known as minor capsid proteins) and (iv) 16 copies of the pp150 molecules [1,18,22]. Three pp150 proteins cluster above each triplex and extend towards the three SCPs atop nearby MCPs, forming a net-like layer of tegument densities that enmesh HCMV capsids [22,24]. The atomic model construct for the N-terminal third of pp150 (pp150nt; residues 1 to 285) suggests that N-terminal residues 1 to 275 alone are sufficient for pp150-capsid binding [22]. Several conserved regions in the N-terminal 275 residues have been identified, including a 27 amino acid cysteine tetrad region, which is conserved across all primate cytomegaloviruses, and two betaherpesvirus-conserved regions (CR1 and CR2) [11,12,13]. The cysteine tetrads and CR1 lie in pp150nt’s upper helix bundle, whereas CR2 lies in the lower helix bundle (Figure 1). Pp150nt–capsid interactions occur in both upper and lower helix bundles and stabilize the nucleocapsid for the production of infectious HCMV virion [18,22].

Peptide therapeutics are a promising new strategy for targeted therapy. Various studies have shown the approach of using peptides as potential antivirals [25,26]. Notable examples of peptide therapeutics against herpesvirus infections include anti-heparan sulfate (HS) peptides [27,28,29,30]. HS and its highly modified form 3-*O*-sulfated HS (3-*O*S HS) is critical for herpes simplex virus-1 (HSV-1) and CMV entry into cells [27,31], and HS removal is required when herpesviruses including CMV is released from the cells after virion maturation [32]. Twelve-mer peptides (G1 and G2) derived from a random M-13 phage display library specifically bind to HS and 3-*O*S HS, and block HSV-1 entry into human corneal fibroblasts and CHO-K1 cells in vitro. In vivo administration of these peptides completely blocks HSV-1 spread in the mouse cornea. G2 peptides isolated against 3-*O*S HS inhibit CMV entry into retinal pigment epithelial cells and thereby block CMV infection [27]. A novel synthetic polybasic peptide (p5 + 14) that binds to hypersulfated HS, effectively inhibits HS-mediated entry of MCMV, HCMV and HSV in vitro [29]. Additionally, the D-form of a p5 + 14 related peptide, known as p5R_D_ effectively reduces CMV infection in vitro and in vivo [30]. All of the above findings indicate that peptides could be used as novel therapeutics against CMV infection.

We targeted the conserved pp150nt regions with sequence-specific peptides, with the goal of developing these peptides into highly effective antivirals. The atomic details of pp150nt structure and its binding interface with capsid proteins (Figure 1A–C) guided the design of peptides targeting CR1, CR2 and cysteine tetrad regions (Table 1; Figure 2). The data show that at least one of these peptides (pep-CR2) is effective in inhibiting HCMV and MCMV growth in cell culture. Microscopic images also suggests that pep-CR2 treated HCMV-infected cells sequester pp150 in the nucleus, thereby compromising the organization of vAC and virion maturation process. Overall, the results in this study indicate that CR2 of pp150 is amenable to targeting by a peptide inhibitor, which can be developed into an effective antiviral.

## 2. Materials and Methods

*Preparation of peptides*: Peptides were designed with a sequence identical to the amino acid sequences in the conserved regions in pp150nt (Table 1). All peptides were modified to include a N-terminal myristoyl group for better cell penetration, and were synthesized by GenScript, USA to >85% purity. Seven mg of lyophilized peptides (MW = 1558.02 daltons) was resuspended in 449.3 µL of 100% dimethyl sulfoxide (DMSO) to achieve a stock concentration of 10 mM before use; however, the working concentration of the peptides included less than 0.1% DMSO.

*Cells*: Human foreskin fibroblasts (HFF) and mouse endothelial fibroblasts (MEF) were cultured in Dulbecco’s modified Eagle’s medium (DMEM; Corning, Manassas, VA; catalog# 10-013-CM) containing 10% fetal bovine serum (Gibco, Life Technologies, Grand Island, NY, catalog# 10437-028), 4.5 g/mL glucose, 2 mM L-glutamine, 1 mM sodium pyruvate, and 100 U/mL penicillin–streptomycin (Corning, Manassas, VA; catalog# 30-002-CI) at 37 °C with 5% CO_2_.

*Virus*: HCMV (TowneBAC and BAD32 strains) was grown on HFF cells and MCMV (K181 strain) was grown on MEF cells. Virus stock was prepared in 3X autoclaved milk, sonicated 3 times for 10 s with a 30 s gap, and stored at −80 °C. The 3X autoclaved milk is prepared from Carnation (Nestle) instant nonfat dry milk powder. Further, 10% milk was prepared in nanopure water, pH was adjusted to 7.0 and it was autoclaved three times.

*Virus infection*: During infection, media was removed from the wells of cell culture plates and appropriately diluted virus stock was absorbed onto the cells in DMEM without serum. Cells were incubated for 1 h with gentle shaking every 10 min followed by washing 3X with serum-free DMEM. Fresh complete medium was added, and cells were incubated until the endpoint.

*Virus titers*: HCMV- and MCMV-infected or mock-infected samples (in triplicates) were harvested within the medium at the designated time points post infection, and stored at −80 °C before titration. On the day of titration, harvested samples (in triplicates) were sonicated three times for 10 s each with 30 s gap. Monolayers of cells (HFFs and MEFs for HCMV and MCMV infection, respectively) were grown in 12-well tissue-culture plates, and serial dilutions of sonicated samples were absorbed onto them for 1 h (in duplicates), followed by 3X washing with serum-free DMEM. For HCMV titers, fresh DMEM containing 10% FBS was added to HFFs and cells were incubated for 9 to 10 days post infection (dpi). For MCMV titers, a carboxymethyl cellulose (CMC; EMD Millipore; catalog# 217274) overlay with complete DMEM (one-part autoclaved CMC and three parts media) was added, and the cells were incubated for 5 days. At the endpoint, DMEM/CMC+DMEM overlay was removed, and cells were washed 2X with PBS. Infected monolayers were fixed in 100% methanol for 5 min. HFFs were immediately stained with Giemsa stain, modified (Sigma-Aldrich, MilliporeSigma, US catalog# GS1L), and MEFs with 1:20 dilution of 1% crystal violet (Fisher Chemicals, Fair Lawn, NJ; catalog# C581-25) for 15 minutes. Plates were finally washed with tap water, air-dried, and plaques with clear zone (MCMV) or dark Giemsa stain (HCMV) were quantified.

*IC_50_ assay and percent inhibition calculation*: To determine the half-maximal inhibitory concentration (IC_50_) of pep-CR2, HFFs were pretreated for 1 h with a range of pep-CR2 concentrations (1 µM, 2.5 µM, 5 µM, 7.5 µM, 10 µM, 12.5 µM, 15 µM, 17.5 µM and 20 µM), pep-control, ganciclovir (GCV) or mock (in triplicates), and then infected with HCMV at a multiplicity of infection (MOI) of 0.1. Cells were fixed at 10 dpi, and the number of foci for each concentration in triplicate wells were enumerated. The data (# of foci) were normalized in GraphPad Prism v9.0, taking the highest number of average foci in the assay (164.66) as 0% inhibition and the lowest number of average foci (2) as 100% inhibition. The percentage (%) inhibition of pep-CR2 was plotted against a concentration range of 1 µM-20 µM on a bar graph. The calculation used in GraphPad Prism was ((value—minOfValues)/(maxOfValues—minOfValues))*100.

*Microscopy*: Samples were prepared using established protocols for fluorescence microscopy. Briefly, HFF cells were grown on coverslip inserts in 24-well tissue-culture plates. Cells were pretreated with pep-control and pep-CR2 at 10 µM concentration for 1 h before infection with HCMV (BAD32 strain where pp150 is GFP tagged) at an MOI of 3.0. At the endpoint (4 dpi), cells were fixed in 3.7% formaldehyde for 10 min and were incubated in 50 mM NH_4_Cl in PBS for 10 min to reduce autofluorescence. Followed by 2X PBS wash, cells were incubated in 0.5% Triton X-100 for 20 min for permeabilization and washed with PBS. Finally, cells were incubated in Hoechst solution (ThermoFisher Scientific, Waltham, MA, USA, Catalog# 33342) in PBS (1:3000) for 10 min to stain the nucleus, followed by PBS wash. Coverslips were retrieved from the wells and were mounted on glass slides with a drop of mounting medium (2.5% DABCO in Fluoromont G), and air-dried for two hours before imaging. Images were acquired on an EVOS-FL epifluorescent microscope (ThermoFisher Scientific, Waltham, MA, USA).

*Cell viability*: HFF and MEF cells were plated in 24-well tissue-culture plates and grown to confluency (in triplicates). Cells were pretreated for 1 h with appropriate concentration of controls and test peptides and then infected with HCMV or MCMV at an MOI of 3.0 or mock-infected. At 3 dpi, HFFs received fresh, complete DMEM containing fresh peptides or controls. The medium was removed at 5 dpi, and HFFs were harvested by trypsinization. For MEFs, CMC+DMEM overlay was removed at 3 dpi before harvesting the cells by trypsinization. Cell viability was determined using trypan blue exclusion using a TC20 automated cell counter (BioRad Laboratories, Hercules, CA, USA) following the manufacturer’s protocol.

*Statistics*: Student’s t-tests were conducted in Graphpad Prism, comparing the means of different groups (GraphPad Prism version 9.0.0, GraphPad Software, San Diego, CA, USA, www.graphpad.com; access date 10 November 2021). Standard error of mean was plotted as error bars. A *p* value of <0.05 was considered significant. An asterisk (*) indicates significant inhibition compared to the wild type. For multiple comparisons, data were analyzed with one-way ANOVA in GraphPad Prism 9.0 and differences between groups were considered significant at a *p* value of <0.05.

## 3. Results

### 3.1. Inhibition of Virus Growth upon Pep-CR2 Treatment

First, we sought to establish the inhibitory efficiency of the peptides. Primary human foreskin fibroblasts (HFF) were pretreated with pep-CR1, pep-CR2, pep-CysTetrad1, pep-CysTetrad2, pep-control and DMSO at 10 µM concentration for 1 h, before infecting them with HCMV at an MOI of 3.0. Peptide concentrations were maintained in the infected cell culture. Cells were harvested at 5 days post infection (dpi), and viral plaque-forming units were enumerated by plating of serial dilutions on fibroblasts. The titers indicated >10-fold reduction in virus growth in pep-CR2-treated cells but not when treated with pep-CR1, pep-CysTetrad1, pep-CysTetrad2, pep-control or DMSO (Figure 3A). To assess the impact of pep-CR2 on virus spread, plaque sizes were calculated and compared between mock vs. pep-CR2-treated groups. Pep-CR2-treated infected cells showed significantly smaller plaque sizes compared to the mock-treated group, indicating reduction in virus spread to the adjacent cells in a foci, hence inhibiting virus growth (Figure 3B,C). Next, to determine the half-maximal inhibitory concentration (IC_50_) of pep-CR2, HFFs were pretreated with pep-CR2 (with a concentration range of 1–20 µM), pep-control and ganciclovir (GCV; as an inhibitory control). Cells were then infected with HCMV at an MOI of 0.1, and virus yield was measured at 10 dpi by enumerating the plaque-forming units. The percentage inhibition of HCMV by pep-CR2 was plotted against the treatment concentration (Figure 4A), and the IC_50_ was calculated to be 1.33 µM with a 95% confidence interval of 1.182 to 1.474 µM. Compared to the pep-control, pep-CR2 showed 90% inhibition of virus titers at 10 µM concentration (Figure 4B), confirming previous results at higher MOI (Figure 3A). To test the cytotoxicity of pep-CR2, a cell viability assay was performed on pep-CR2 (10 µM)-treated infected and uninfected HFFs along with appropriate controls. Cell viability was similar across treatment groups in this assay (Figure 5A). Furthermore, pep-CR2-treated infected cells had significantly higher cell viability compared to the control groups, indicating that pep-CR2 efficiently protected the HCMV-infected cells from virus-induced lytic death (Figure 5B). Overall, the data from these experiments indicate a significant reduction in virus growth upon pep-CR2 treatments, without impacting cell viability.

### 3.2. Pep-CR2 Treatment Sequesters pp150 in the Nucleus of Infected Cells

To study the localization of pp150 in pep-CR2- and pep-control-treated infected cells, HFFs were pretreated with pep-CR2 or pep-control, as described earlier, and then infected with a strain of HCMV where pp150 is fused with GFP (BAD32, MOI 3.0) [20]. Cells were fixed and imaged by epifluorescent microscopy at 4 dpi. These localization studies showed that pep-CR2-treated cells contain a diffused GFP signal in the cytoplasm of infected cells in contrast to a solid juxtanuclear sphere of GFP, which corresponds to vAC in the pep-control-treated cells (Figure 6). The intensity of the GFP signal was much higher in the nucleus of pep-CR2-treated cells compared to the nucleus of the pep-control-treated cells. These results indicate that pep-CR2 may be compromising virion maturation by sequestering pp150 in the nucleus of infected cells and interfering with the organization of the vAC.

### 3.3. Pep-CR2 Shows Similar Inhibitory Potential against Murine Cytomegalovirus (MCMV)

Conserved virion tegument protein pp150 is encoded by HCMV UL32 gene. The amino terminal one-third of pp150 contains CR1 and CR2, which are known to be the two most highly conserved regions among all betaherpesvirus pp150s [13]. To further strengthen our hypothesis that targeting of conserved regions of pp150 by pep-CR2 blocks CMV infection, we investigated the inhibitory potential of pep-CR2 against MCMV. Since CMV is species specific, we performed the MCMV experiments in mouse endothelial fibroblasts (MEFs). MEFs were pretreated with pep-CR2 and pep-control, as described earlier, before infecting them with MCMV (K181) at an MOI of 3.0. Peptide concentrations were maintained in the infected cell culture. Cells were harvested at 3 dpi, and virus titers were enumerated. The titers indicate a >10-fold reduction in virus growth in pep-CR2-treated cells compared with the pep-control treatment group (Figure 7A). Next, to rule out any cytotoxic effects of pep-CR2, a cell viability assay was performed on pep-CR2-treated infected and uninfected MEFs along with pep-control and GCV as appropriate controls. Cell viability was not affected at the treated concentration of pep-CR2 in uninfected cells (Figure 7B). Additionally, pep-CR2-treated infected cells had higher cell viability compared with control groups, indicating that pep-CR2 efficiently protected the MCMV-infected cells from virus-induced lytic death (Figure 7C). Overall, these results indicate that pep-CR2 shows similar inhibitory potential against MCMV compared to HCMV.

## 4. Discussion

In this study, we utilized multiple approaches to demonstrate that CR2 of CMV tegument protein pp150 is amenable to targeting by sequence-specific peptide pep-CR2. We first screened the inhibitory efficiency of peptides that targeted the conserved regions in pp150. The results indicated that upon treatment with peptides that target CR2 of pp150 (pep-CR2), there is a significant reduction in virus growth in cell culture (Figure 3). To further investigate the dose response of pep-CR2 on infected human fibroblasts, we performed an IC_50_ assay using a concentration range of 1 to 20 µM. The half-maximal inhibitory concentration was calculated to be 1.33 µM (Figure 4A), and pep-CR2 showed ~90% inhibition of virus growth at 10 µM concentration (Figure 4B). Pep-CR2 treatment of HFFs did not affect cell viability for the duration of treatment (Figure 5A) confirming that the observed reduction in virus titer was not due to cell death. Moreover, when HFs were pretreated and maintained in pep-CR2 throughout the course of infection, the cells resisted infection-induced cell death at late time post infection (Figure 5B). To understand the possible mechanism of inhibition of HCMV replication by pep-CR2, we looked at the cytoplasmic and nuclear localization of pp150 at late time post infection. The results indicated that pp150 is sequestered in the nucleus of infected cells that are treated with pep-CR2 (Figure 6). This would conform to a mechanistic disruption of pp150 loading onto capsids and subsequent cytoplasmic egress, which might compromise vAC organization and virion maturation processes.

Tegument proteins play a critical role in the CMV life cycle process in the host cell, and can be considered as important targets for the development of antiviral therapy. Pp150 is a late viral tegument protein which supports the final steps of the maturation process [9,11,12,13]. CMV infection can persist in latent, chronic and productive stages depending on the host physiology and the types of cells being infected [1]. CMV most likely establishes life-long persistence by replicating at a subclinical level and occasionally reactivating from the latent stage triggered by various biological, chemical or physical stimulations including immunological disorders, radiation and cell differentiation among others [1,33,34]. Therefore, an effective antiviral against CMV would target cell-to-cell spread of latently reactivated virus. This can be effectively achieved by targeting late stages of virus maturation.

During the primary CMV virion maturation, capsid assembly and DNA encapsidation take place inside the nucleus [1]. After the nucleocapsid is formed, it acquires some tegument proteins, including pp150, and migrates from the nucleus to the cytoplasm. Once the nucleocapsid is in the cytoplasm, it accumulates in a ring-shaped perinuclear viral assembly compartment (vAC), where the final or secondary steps of maturation take place. Pp150 is a large, conserved, CMV virion tegument protein, which hitches a ride on the capsids during the nuclear egress and accumulates in the vAC [8,9,11,12]. Pp150 is well-characterized to control cytoplasmic events during virus maturation as it stabilizes and retains the nucleocapsid organization throughout the final envelopment inside the vAC [12].

Pep-CR2 likely interferes with the pp150–capsid interaction in the nucleus of the cells, thereby blocking pp150 loading onto capsids and its translocation out of the nucleus and into the cytoplasm. Additionally, viral DNA can exit the nucleus only when is it packaged inside the capsid. In the absence of pp150, or if the pp150–capsid interaction is abolished, the nucleocapsids can still exit the nucleus but most likely will be degraded in the cytoplasm [12]. Pep-CR2 would block pp150 from capsid binding by preventing interactions with the triplexes. This interface may not be completely evident in the 2D models depicted in Figure 1; however, conformational changes may occur that could allow CR2 and the triplexes to come closer together. Alternatively, pep-CR2 could be binding to a capsid protein prior to their nuclear localization or capsid assembly that then would prevent pp150 association. Furthermore, the peptide could be preventing other host or viral protein interactions with pp150 that would also result in a similar phenotype to what is observed in this study.

Pp150-interacting host proteins determine virus maturation, and a disruption of these interactions can compromise the virus maturation process. Pp150 interacts with Bicaudal D1 (BicD1), a protein that regulates trafficking within the secretory pathway, and this interaction is required for the correct localization of pp150 in the vAC of the infected cells [19]. In the presence of pep-CR2, the BicD1–pp150 interaction will be blocked or reduced, since the exit of pp150 from the nucleus to the cytoplasm is affected. This will interfere with the formation of vAC and proper virus maturation. Other CMV proteins such as pUL96 interact with pp150 and stabilize pp150-associated nucleocapsids [14]. Interfering with the pp150–pUL96 interaction will also lead to a defect in the formation of vAC, since pUL96 and pp150 are both implicated in the formation and maintenance of vAC.

Additionally, dynamin–clathrin-mediated endocytic pathways are important for cytoplasmic stages of CMV maturation [35]. Pp150 is known to interact directly with clathrin, which affects the virion maturation process [20]. In the presence of pep-CR2, the pp150–clathrin interaction would also be compromised due to reduced availability of pp150 in the cytoplasm, which in turn would affect the vAC organization and the maturation process. BicD1 also interacts with the dynamin–clathrin-mediated membrane trafficking pathway [36]. Since pp150 is known to interact with both BicD1 and clathrin, it is possible that the presence of pep-CR2 would disrupt pp150–clathrin and/or pp150–BicD and/or BicD1–clathrin interactions, resulting in impaired trafficking and virus maturation.

Cryo-electron microscopic studies on isolated B capsids purified from HCMV- and simian cytomegalovirus (SCMV)-infected cells show that pp150 interacts with capsid proteins during virus maturation [37,38]. The amino terminal one-third of HCMV pp150 (pp150nt) as well as the SCMV homolog of UL32 are responsible for capsid binding [13]. This pp150nt–capsid interaction is important for the virion maturation process because it retains and stabilizes the organization of nucleocapsids throughout the final steps of maturation [1,12,18]. Earlier studies using a HCMV UL32-deletion mutant showed that pp150 is critical for virion egress [11]. However, subsequent transmission-electron-microscopic studies of UL32-deletion-mutant (ΔUL32)-infected cells revealed that although nucleocapsids in the nucleus appeared similar in both wild-type and ΔUL32-mutant cells, there were fewer cytoplasmic virus particles in mutant cells compared to WT, indicating that in the absence of pp150, maturing nucleocapsids are unstable in the cytoplasm following nuclear egress [12].

Pp150nt (amino acid residues 1–275) is necessary and sufficient for capsid–tegument interaction, and it consists of conserved regions including CR1 and CR2. Deletion of the amino terminal of pp150 or disruption of either CR1 or CR2 in SCMV blocked the secondary spread of mutant virus, thereby indicating that CR1 and CR2 domains are critical for virus maturation [11]. Further studies by Tandon and Mocarski successfully characterized CR2 phenotypes, where they showed that two independent CR2 point mutants failed to support viral replication. These replication-defective CR2 mutants exhibited a phenotype similar to a virus that carried a complete deletion of UL32 ORF (ΔUL32). The UL32 mutant-virus-infected cells showed defect in the formation of vAC, which was highly vesiculated and contained fewer nucleocapsids or complete virus particles in contrast with the intact, wild-type vAC [12]. The pp150 localization results in HCMV-infected fibroblasts (Figure 6) are consistent with these earlier findings, where the perturbation of pp150-CR2 by peptides (pep-CR2) compromises the formation of vAC, thereby affecting virus maturation.

To further strengthen our hypothesis that CR2 of pp150 can be targeted by pep-CR2, we investigated the inhibitory efficiency of pep-CR2 against MCMV. CR1 and CR2 are the two most highly conserved regions among various CMV species, and it is expected that pep-CR2 will target CR2 of M32 (UL32 homolog of pp150 in MCMV). The results in Figure 7 show that pep-CR2 treatment significantly reduces MCMV growth with no impact on cell viability, which corroborates the results of pep-CR2 treatment on HCMV-infected cells.

Overall, the results in this study show that CR2 of pp150 is amenable to targeting by a sequence-specific peptide inhibitor, which can be developed into an effective antiviral.

## Figures and Tables

**Figure 1 viruses-13-02277-f001:**
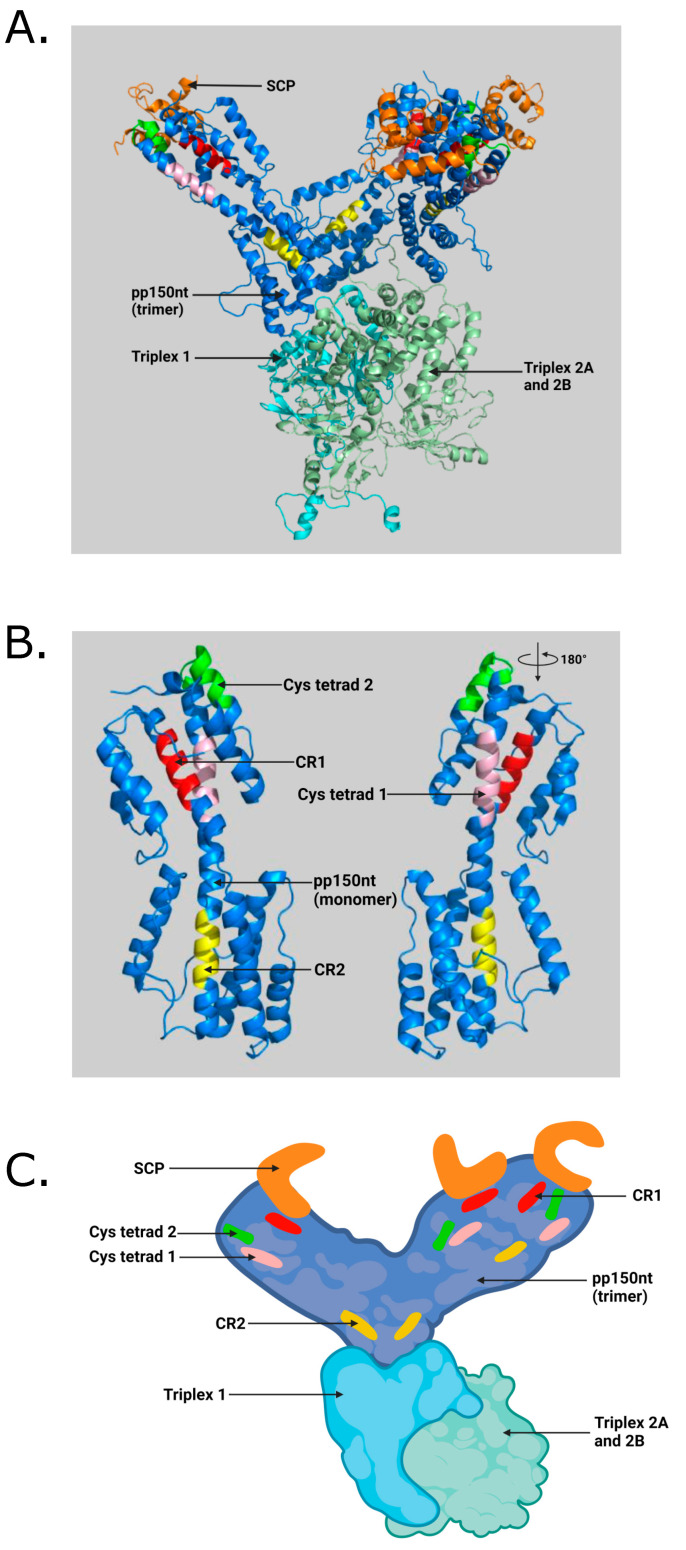
(**A**) Illustration of pp150nt structure and its binding interface with capsid proteins in an HCMV virion. SCP (small capsid protein) binds to MCP (major capsid protein). Triplexes (also known as minor capsid-binding protein) are heterotrimers composed of Tri1 and a Tri2A–Tri2B heterodimer. Three pp150nt residues (pp150nt trimer) cluster above triplexes and extend towards SCP atop nearby MCP, forming a net-like layer of tegument densities. (**B**) Close-up view showing conserved regions in pp150 monomer generated by PyMOL software (Schrödinger, LLC) using RSCB PDB ID: 5VKU and UNIPROT ID: Q6SW99. (**C**) Cartoon illustration of pp150nt trimer and its binding interface with capsid proteins in an HCMV virion generated with BioRender.com (access date: 10 November 2021) Orange—SCP; Blue—pp150nt; Cyan—Triplex 1; Pale green—Triplex 2A and 2B; Red—pp150 conserved region 1 (CR1); Yellow—pp150 conserved region 2 (CR2); Pink—Cysteine tetrad 1; Green—Cysteine tetrad 2 [22].

**Figure 2 viruses-13-02277-f002:**
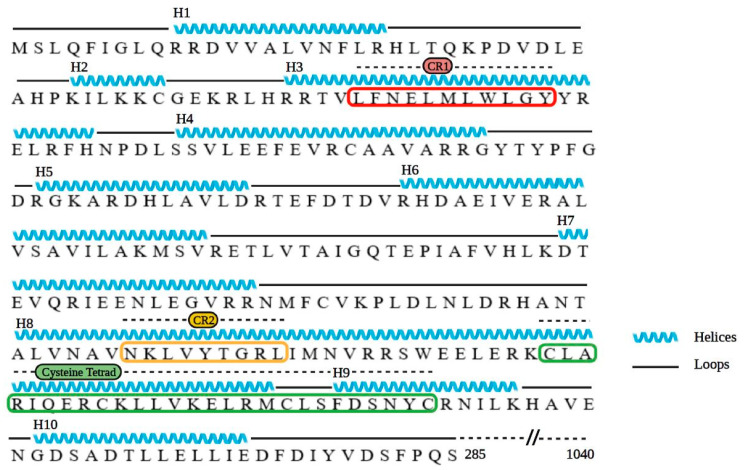
Amino acid sequence and structure-derived secondary structure of pp150nt. Black lines represent loops, blue lines represent helices, CR1 is boxed in red, CR2 boxed in yellow and cysteine tetrad boxed in green. Illustrated in BioRender.com (access date: 10 November 2021) using RSCB PDB ID: 5VKU and UNIPROT ID: Q6SW99 [22].

**Figure 3 viruses-13-02277-f003:**
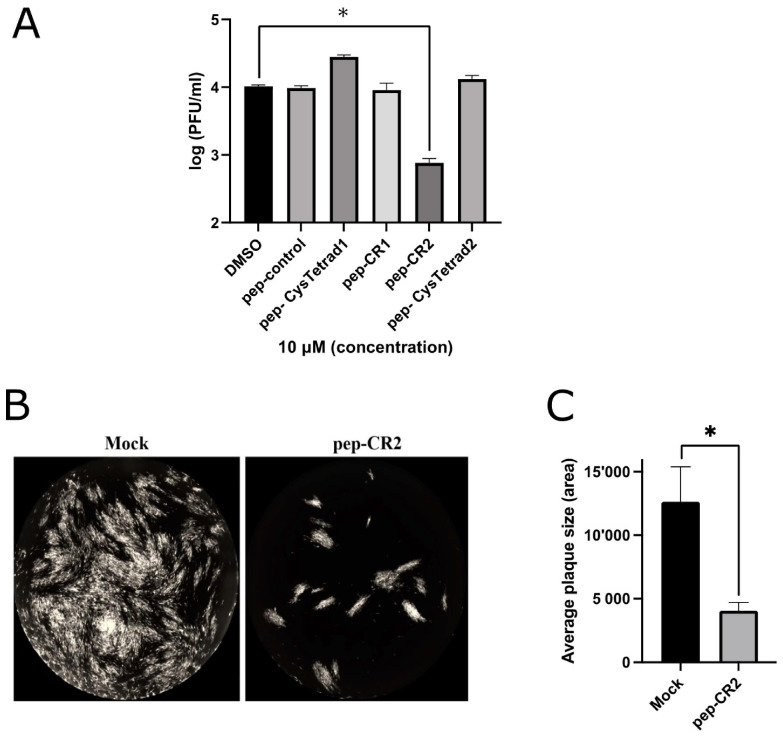
(**A**) Inhibition of virus growth upon pep-CR2 treatment. Cells (human foreskin fibroblasts (HFF)) were pretreated with 10 µM concentration of control (DMSO), pep-control, pep-CysTetrad1, pep-CysTetrad2, pep-CR1 and pep-CR2 (in triplicates) and infected with HCMV at an MOI of 3.0. Cells were harvested at 5 dpi (days post infection) and virus titers were assessed on HFFs. Data were analyzed by Student’s *t*-tests, comparing the means of DMSO control and the peptide-treated group. (**B**) GFP+ images showing plaque size comparison in pep-CR2 and mock-treatment control group. Cells were pretreated with pep-CR2 (10 µM) and mock for 1 h (in triplicates) before infecting them with HCMV at a low MOI of 0.1. Cells were fixed at 10 dpi and virus yield was measured by counting the number of plaques. Images were acquired on a Cytation5 (BioTek Instruments, USA) cell-imaging reader with 4X magnification. (**C**) Plaque size comparison between pep-CR2 and mock-treated cells. Individual plaque sizes were measured by counting the area of the plaques in ImageJ software (National Institutes of Health, Bethesda, MD, USA). Standard error of mean was plotted as error bars. Data were analyzed by Student’s t-tests, comparing the means of the test and the control group. A *p* value of <0.05 was considered significant. An asterisk (*) indicates significant inhibition compared to wild type.

**Figure 4 viruses-13-02277-f004:**
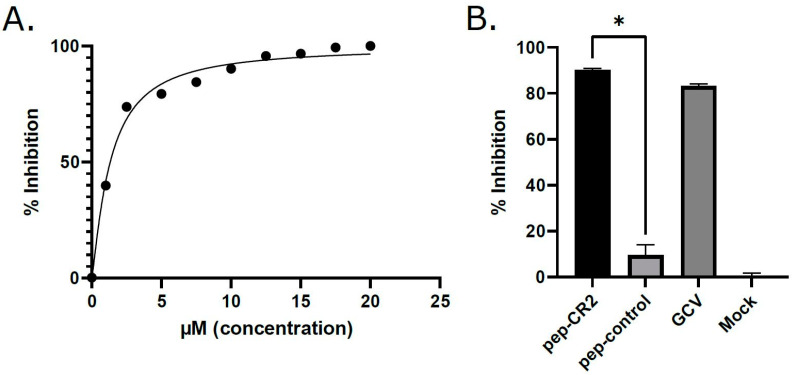
(**A**) Determining the IC_50_ (half-maximal inhibitory concentration) of pep-CR2. HFFs were pretreated for 1 h with different concentrations of pep-CR2 (1 µM, 2.5 µM, 5 µM, 7.5 µM, 10 µM, 12.5 µM, 15 µM, 17.5 µM and 20 µM), pep-control, ganciclovir (GCV) or mock (in triplicates) and then infected with HCMV at a low MOI of 0.1. Cells were fixed at 10 dpi, and virus yield was measured by counting the number of plaques. The percentage inhibition of pep-CR2 is plotted against concentration range of 1 µM-20 µM. The IC_50_ of pep-CR2 was calculated to be 1.33 µM with a 95% confidence interval of 1.182 to 1.474 µM. (**B**) Comparing inhibition of virus growth by pep-CR2, pep-control, GCV and no-treatment control (Mock) at 10 µM concentration. Pep-CR2 treated cells showed ~90% inhibition in virus growth compared to the controls. Data were analyzed by Student’s *t*-tests, comparing the means of control and treatment groups. Standard error of mean was plotted as error bars. A *p* value of <0.05 was considered significant. An asterisk (*) indicates significant inhibition compared to wild type.

**Figure 5 viruses-13-02277-f005:**
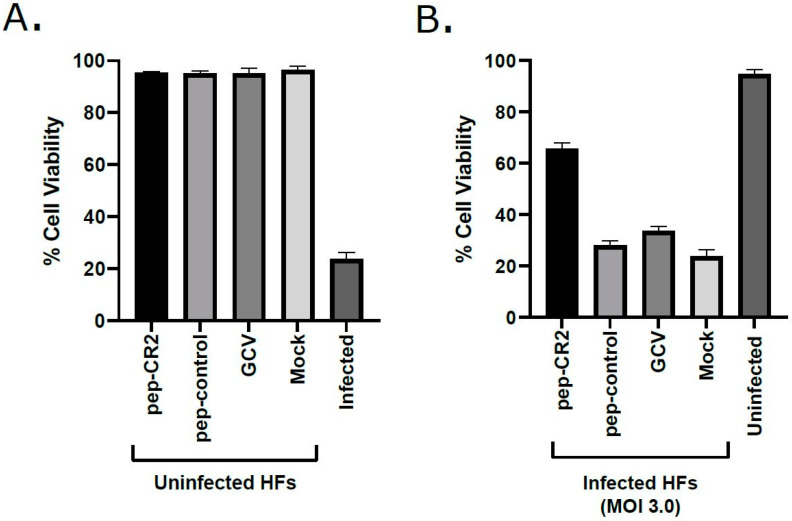
Cell viability (%) in (**A**) pep-CR2-treated uninfected cells vs. (**B**) pep-CR2-treated infected cells. Cells were pretreated with pep-CR2 as well as with appropriate controls (pep-control and GCV) or mock (in triplicates) for 1 h and were then either infected with HCMV at a high MOI of 3.0 or mock-infected. Cell viability was performed by trypan blue exclusion assay at 5 dpi for both groups. Results indicate that pep-CR2 protects cells from virus-induced lytic cell death. Samples were analyzed with one-way ANOVA in GraphPad Prism 9.0 and showed significant differences between groups (*p* < 0.05).

**Figure 6 viruses-13-02277-f006:**
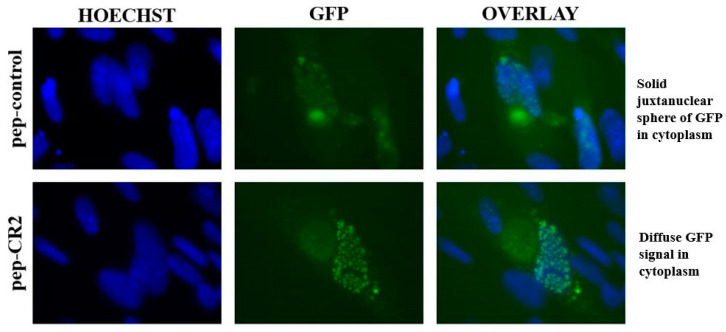
Epifluorescent imaging indicating pp150 localization in the nucleus and the cytoplasm of pep-CR2 and pep-control-treated cells. Cells were pretreated with pep-control and pep-CR2 at 10 µM concentration for 1 h before infecting them with HCMV (BAD32 strain where pp150 is GFP tagged) at an MOI of 3.0. Cells were fixed and imaged under an epifluorescent microscope at 4 dpi. Hoechst stains the nucleus. Results indicate that pep-CR2 may be compromising virion maturation by sequestering pp150 in the nucleus of infected cells and interfering with the organization of the virus assembly compartment (vAC).

**Figure 7 viruses-13-02277-f007:**
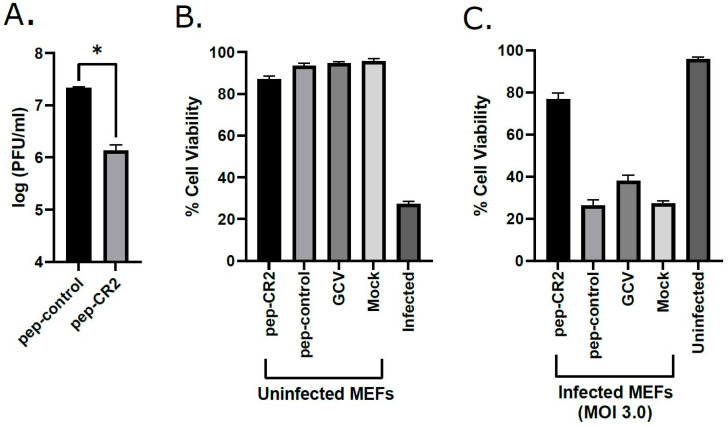
(**A**) Inhibition of MCMV growth upon pep-CR2 treatment. MEF cells were pretreated with 10 µM concentration of pep-CR2 and pep-control (in triplicates) and infected with MCMV at an MOI of 3.0. Cells were harvested at 3 dpi and virus titers were assessed on MEFs. Data were analyzed by Student’s t-tests, comparing the means of pep-control and pep-CR2. Standard error of mean was plotted as error bars. A *p* value of <0.05 was considered significant. An asterisk (*) indicates significant inhibition compared to wild type. Cell viability (%) in (**B**) pep-CR2-treated uninfected MEFs vs. (**C**) pep-CR2-treated infected MEFs. Cells were pretreated with pep-CR2 as well as with appropriate controls (pep-control and GCV) or mock (in triplicates) for 1 h and were then either infected with MCMV at a high MOI of 3.0 or mock-infected. Cell viability was performed by trypan blue exclusion assay at 3 dpi for both groups. Results indicate that pep-CR2 protects cells from virus-induced lytic cell death. Samples were analyzed with one-way ANOVA in GraphPad Prism 9.0 and showed significant differences between groups (*p* < 0.05).

**Table 1 viruses-13-02277-t001:** Amino acid sequence of test and control peptides ^1^.

Peptide	Sequence	HCMV Pp150 UNIPROT Entry (Q6SW99) Residue Numbers
Pep-control	DYKDDDDK (Flag Sequence) Control	NA
Pep-CR1	LFNELMLWL (CR1)	52–60
Pep-CR2	NKLVYTGRL (CR2)	201–209
Pep-CysTetrad1	KCLARIQERCK (Cysteine Tetrad 1)	223–233
Pep-CysTetrad2	MCLSFDSNYCR (Cysteine Tetrad 2)	241–251

^1^ All peptides were synthesized with a N-terminal myristoylation by GenScript, USA.

## Data Availability

All data are included in the manuscript itself.

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
