# Peer review of "Tegument Protein pp150 Sequence-Specific Peptide Blocks Cytomegalovirus Infection"

_viruses, 2021, doi:10.3390/v13112277_

Round 1

Reviewer 1 Report

This is an interesting article from The Tandon laboratory on the identification of peptide mimics that inhibit human cytomegalovirus replication. This protein is an essential component of the regiment structure and is important for virus egress and maturation. The authors use high resolution images of the capsid-tegument structure of HCMV to derive a number of peptides that comprise conserved regions (helical in structure) and a cysteine tetrad domain. The peptides were used in cell culture assays to demonstrate significant (10 fold) inhibition by a peptide spanning the CR2 sequence. They show this peptide has comparable inhibitory activity against HCMV as gangcyclovir. They also show that the peptide may disrupt correct localization of p150 in the cytoplasm. Finally they couple the peptide to a moeity that acts to stabilize these peptides (potentially for use in a therapeutic model) and show the peptide inhibits plaque size.

This is a well thought designed project which has practical applications. The experiments are carefully designed and the resulting data are carefully analyzed and presented. This is the correct venue for this study and it should be of great interest to the community that investigates therapeutic interventions against herpesviruses.

Comments:

A more sophisticated viability assay maybe better appreciated by the reader. Also viability assays might better be placed in the main text instead of the supplemental figures.

The ELP moiety is large (approximately 50 kD). Does this not defeat the goal of using small peptide mimetics for inhibition? It also has the added  complication of having anti-HCMV properties by itself. Although the authors conclude that the inhibition with the peptide conjugated and peptide alone are comparable, the biological assays used (plaque assay ELP-peptide versus growth curve - peptide) are very different. The authors need to address these issues. Some of these points are addressed in the discussion and potential alternatives.

It maybe more useful to discuss how the peptide may inhibit correct localization and virus replication. Cryo analysis predicts interactions with the small capsid protein as well as the triplex proteins. Or other potential structural protein interactions that could account for the localization phenotype.

Author Response

Review #1

Comments and Suggestions for Authors

This is an interesting article from The Tandon laboratory on the identification of peptide mimics that inhibit human cytomegalovirus replication. This protein is an essential component of the regiment structure and is important for virus egress and maturation. The authors use high-resolution images of the capsid-tegument structure of HCMV to derive a number of peptides that comprise conserved regions (helical in structure) and a cysteine tetrad domain. The peptides were used in cell culture assays to demonstrate significant (10-fold) inhibition by a peptide spanning the CR2 sequence. They show this peptide has comparable inhibitory activity against HCMV as ganciclovir moiety. They also show that the peptide may disrupt the correct localization of p150 in the cytoplasm. Finally, they couple the peptide to a moiety that acts to stabilize these peptides (potentially for use in a therapeutic model) and show the peptide inhibits plaque size.

This is a well-thought-designed project which has practical applications. The experiments are carefully designed and the resulting data are carefully analyzed and presented. This is the correct venue for this study and it should be of great interest to the community that investigates therapeutic interventions against herpesviruses.

Comments:

  1. A more sophisticated viability assay may be better appreciated by the reader. Also, viability assays might better be placed in the main text instead of the supplemental figures.

We have moved the cell viability assay figures from the supplemental materials to the main manuscript (Fig. 5). We also investigated the effect of pep-CR2 on MCMV infection in mouse endothelial fibroblasts and added cell viability data on pep-CR2 treated MEFs in the manuscript (Fig. 7).

Trypan blue exclusion assay is widely accepted and very commonly used to measure cell viability. However, in order to assess the cell viability using a more refined method, we have used ATP quantification. In this luminescence-based cell viability assay, the number of viable cells in culture is detected by quantifying ATPs, which indicates the presence of metabolically active cells. In this assay the cells are exposed to the enzyme luciferase and the substrate luciferin. The first step of the luciferase reaction is the adenylation of the substrate luciferin which utilizes ATP, and the second step is the oxidative decarboxylation of the luciferyl adenylate to form oxyluciferin. The oxyluciferin formed is at an excited state, and light is produced when it moves back to the ground state. Since the luciferase reaction relies on ATP, only living cells in the culture would produce a luminescent signal.

We pretreated human foreskin fibroblasts (HFF) with pep-CR2, pep-control and ganciclovir (GCV) and measured the ATP using CellTiter-Glo Luminescent Cell Viability Assay Kit (Promega Corporation, USA; catalog# G7570) five days post-treatment. The results indicate high cell viability in all groups.

  1. The ELP moiety is large (approximately 50 kD). Does this not defeat the goal of using small peptide mimetics for inhibition? It also has the added complication of having anti-HCMV properties by itself. Although the authors conclude that the inhibition with the peptide conjugated and peptide alone are comparable, the biological assays used (plaque assay ELP-peptide versus growth curve - peptide) are very different. The authors need to address these issues. Some of these points are addressed in the discussion and potential alternatives.

We have removed ELP studies from this manuscript as they would be suitable for further investigations and future studies.

  1. It may be more useful to discuss how the peptide may inhibit correct localization and virus replication. Cryo analysis predicts interactions with the small capsid protein as well as the triplex proteins. Or other potential structural protein interactions that could account for the localization phenotype.

We have added these details to the discussion. During the primary CMV virion maturation, capsid assembly and DNA encapsidation take place inside the nucleus. After the nucleocapsid (NC) is formed it acquires the tegument protein and migrates from the nucleus to the cytoplasm (nuclear egress). Once the NC is in the cytoplasm it accumulates in a ring-shaped perinuclear viral assembly compartment (vAC) where the final or secondary steps of maturation occurs. Pp150 is a large conserved CMV virion tegument protein, which hitches a ride on the capsids during the nuclear egress and accumulates in the vAC. Pp150 is well characterized to control cytoplasmic events during virion maturation. pp150 stabilizes and retains the nucleocapsid organization throughout the final envelopment inside the vAC.

The localization of capsids and pp150 is affected by pep-CR2. Pep-CR2 likely interferes with the pp150-capsid interaction in the nucleus of the cells thereby blocking pp150 loading onto capsids and translocation of pp150 out of the nucleus to the cytoplasm, which compromises the organization of vAC and virion maturation process. Also, viral DNA can exit the nucleus only when is it packaged inside the capsid. In the absence of pp150, or if the pp150-capsid interaction is abolished, the NC can still exit the nucleus but most likely it will be degraded in the cytoplasm (because pp150 maintains and stabilizes the NC organization) and therefore compromise the virion maturation process.

pp150 interacting host proteins also determine maturation and a disruption of these interactions can compromise the virion maturation process as well. Pp150 interacts with Bicaudal D1 (BicD1), a protein that regulates trafficking within the secretory pathway and this interaction is required for the localization of pp150 in the vAC in infected cells (PMID 20089649). In presence of pep-CR2, the BicD1-pp150 interaction will be blocked or reduced because pep-CR2 would also block the exit of pp150 from the nucleus to the cytoplasm. This will interfere with the formation of vAC and account for a less compact pp150 presence in the cytoplasm.

Other CMV proteins like pUL96 interact with pp150 and stabilize pp150 associated nucleocapsids (PMID 21593167). Interfering with pp150-pUL96 interaction will also lead to a defect in the formation of vAC since pUL96 and pp150 are both implicated in the formation and maintenance of the assembly compartment. Also, dynamin and clathrin are critical for the formation of vAC.

Dynamin-clathrin mediated endocytic pathways are important for cytoplasmic stages of CMV maturation (PMID 30282704). pp150 is known to interact directly with clathrin, which affects the virion maturation process (PMID 20023299). In the presence of pep-CR2, the pp150-clathrin interaction would also be compromised due to reduced availability of pp150 in the cytoplasm, which in turn would affect the vAC organization and maturation process.

BicD1 also interacts with dynamin-clathrin mediated membrane trafficking pathway [PMID 20111007]. Since pp150 is known to interact with both BicD1 and clathrin, it is possible that the presence of pep-CR2 can disrupt pp150-clathrin and/or pp150-BicD and/or BicD1-clathrin interactions.

Reviewer 2 Report

CMV infection can cause congenital defects and is associated with cardiovascular diseases. Better drugs are needed to treat the infection. The authors demonstrate that CR2 of pp150 is can be a viable target to reduce infection by a peptide inhibitor. The peptide therefore, can be developed into an effective antiviral drug candidate. The manuscript is based on an innovative idea and composed of novel findings. For the most part manuscript is well written and easy to understand. A few issues, listed below, need to be resolved before it can be considered for publication.

Introduction: Please cite and discuss how heparan sulfate can facilitate CMV entry and peptides can be used to block heparan sulfate and hence, block CMV infection [PMID: 21596749]. Also, heparan sulfate removal is needed when herpesviruses including CMV mature and released from cells [PMID: 25912399].

Fig. 1B and Table 1, many additional regions are shown but not targeted, please explain.

Fig. 3, please show results at an additional MOI because natural infection does not always occur at a fixed MOI. Using an additional strain will add more credence to the findings.

Fig. 4, Could the IC50 be limited to the initial dose of virus infection? Additional experiments as suggested for Fig. 3 will help.

Fig. 5, Why only one or two cells take up the peptide? Would it not adversely affect the treatment?

Fig. 6, Any evidence that the result is not virus dose specific? How about strain specificity?

Author Response

Review #2

Comments and Suggestions for Authors

CMV infection can cause congenital defects and is associated with cardiovascular diseases. Better drugs are needed to treat the infection. The authors demonstrate that CR2 of pp150 can be a viable target to reduce infection by a peptide inhibitor. The peptide, therefore, can be developed into an effective antiviral drug candidate. The manuscript is based on an innovative idea and is composed of novel findings. For the most part, manuscript is well written and easy to understand. A few issues, listed below, need to be resolved before it can be considered for publication.

  1. Introduction: Please cite and discuss how heparan sulfate can facilitate CMV entry and peptides can be used to block heparan sulfate and hence, block CMV infection [PMID: 21596749]. Also, heparan sulfate removal is needed when herpesviruses including CMV mature and released from cells [PMID: 25912399].

We have added the citations and discussed the role of anti-HS peptides blocking CMV infection in the introduction section (paragraph 5).

  1. 1B and Table 1, many additional regions are shown but not targeted, please explain.

Figure 1 has been replaced by ribbon diagrams generated by PyMOL to clearly illustrate regions important for pp150 interactions and we have also added a cartoon (Fig 1C) to depict the same.

  1. Please show results at an additional MOI because natural infection does not always occur at a fixed MOI. Using an additional strain will add more credence to the findings.

We have shown the peptide inhibition results at a high MOI of 3.0 (Fig. 3A and 7A) and at a low MOI of 0.1 (Fig 3B, 3C, 4A and 4B). In both MOI settings pep-CR2 shows a significant reduction (90% reduction) in virus titers.

We have used two strains of HCMV in this manuscript, Towne-BAC, and BAD32 (AD169 strain). We also showed additional pep-CR2 data on murine cytomegalovirus (MCMV) K181 strain infected mouse endothelial fibroblast cells (MEF) (Fig 7).  

Pp150 is a conserved CMV virion tegument protein. The N terminal one-third of pp150 (pp150nt) has several conserved regions including a cysteine tetrad, which is conserved among all primate CMVs and conserved regions 1 and 2 (CR1 and CR2), which are conserved among all betaherpesviruses. CR2 is present in all CMV strains. Studies by Baxter et. al showed sequence alignments of different CMV species which indicates CR1 and CR2 as the two most highly conserved regions (Fig 2; PMID 11435566).

  1. Could the IC50 be limited to the initial dose of virus infection? Additional experiments as suggested for Fig. 3 will help.

We show the pep-CR2 mediated inhibition of HCMV in both high (3.0) and low (0.1) MOI settings. At 10µM peptide (pep-CR2) concentration, we see 90% reduction in virus titer in high MOI settings, which corroborates with the low MOI IC50 data. At low MOI, all the cells in culture are not infected at the same time that allows for multiple replication cycles in contrast to high MOI infections.

  1. Why only one or two cells take up the peptide? Would it not adversely affect the treatment?

In this figure (now Fig.6) we are imaging the CMV pp150 tegument protein. For this localization assay we used HCMV BAD32 strain where pp150 is GFP tagged. We have not imaged the peptide and the number of cells that were able to intake the peptide as it is technically challenging to image and track the very small peptide.

  1. 6, Any evidence that the result is not virus dose-specific? How about strain specificity?

We have removed ELP studies from this manuscript as they would be suitable for further investigations and future studies.

Reviewer 3 Report

In their manuscript entitled ‚Human cytomegalovirus essential tegument protein pp150 is amenable to targeting by sequence specific peptides and ELP-conjugates’ the authors Mitra et al. investigated the potential of four pp150-derived peptides of between nine to eleven amino acids to inhibit HCMV replication in human foreskin fibroblasts. They observed that in case of one of these peptides, an inhibition with an IC50 of about 1 micromolar can be recorded. Furthermore, they observed that this peptide retains inhibitory activity when an ELP-tail is added to this peptide. However, this fusion reduces the inhibitory activity by a factor of 20-fold. All experiments appear sound and are properly described. The overall scope of the manuscript is however quite narrow.

Major points

In the title, the authors write that pp150 is amendable to targeting by sequence specific peptides and in the discussion, ‘we screen the inhibitory efficiency of peptides that targeted the conserved region of pp150’. I consider that many readers (including myself) expect from the title that these peptides target pp150, namely that they somehow interact with pp150 and thereby block pp150 activity. Even when omitting the interaction expectation, targeting would still be read as pp150 being mechanistically targeted by these compounds. However, this is not the case at all here, since the manuscript investigates pp150-derived peptides, only. Whether these are able to block wild-type pp150 function is not at all investigated in the manuscript. I would therefore strongly suggest to reword the title and the manuscript with respect to this.

Figure 1 is very poor and doesn’t really explain/illustrate anything. I did look up the PDB entry 5VKU on which the illustration is based. I do concede that the structure of an entire virus is difficult to illustrate but if the authors aim to explain how pp150 interacts with the HCMV capsid and more precisely with the proteins MCP, SCP and the triplex proteins then they have to provide a better illustration. My suggestion would be to provide A) a sketch that roughly illustrates that pp150 sits on top of the triplexes and inbetween the MCP/SCP hexameric and pentameric complexes and B) a cartoon/ribbon illustration of pp150 with the areas/segments of pp150 of interest (CR1, CR2, Cys Tetrad 1 and Cys Tetrad 2) highlighted in color.

The labelling of the CR2 peptide with an elastin-like polypeptide (ELP) does not add any important additional information to the manuscript since the labelling is clearly detrimental to the inhibitory activity of the CR2 peptide (greater 20-fold reduction). The authors write that it is known that ELP fusions enhance in-vivo bioavailability. However, no experimental data are provided that investigate such an advantage in case of the CR2 peptide. In the competing interest section, the authors write that one coauthor is involved in a company that works on developing ELP drug delivery technology. Since ELP doesn’t provide any advantage in the context of the present study (on contrary), I would suggest that all experiments related to ELP are removed. Likewise, the authors could provide additional experiments that show a clear benefit for using ELP-CR2 fusion constructs.

Minor points

Table 1: The authors marked the sequences in parentheses with attributes such as ‘resembles CR1’ etc. Since some of these sequences are identical to the wild-type sequences, the word ‘resembles’ appears confusing. I would also suggest that the authors provide in the table the residues numbers of the sequence segments in relation to a defined UNIPROT entry.

Figure 2: The title of the figure reads: ‘Amino acid sequence and predicted secondary structure of pp150nt.’ The wording ‘predicted’ is confusing. In structural biology the word prediction relates to structural information that is not available from experimental data but which needs to be bioinformatically inferred. However, in case of pp150nt, several experimental structures are available, hence no prediction is needed. In would be better to use the wording: ‘... and structure-derived secondary structure...’. The authors also write that Figure 2 was generated with the PyMOL software. Is this correct? I am not aware that Pymol creates such illustrations. However, I might be mistaken here. In relation to sequence information, the authors should provide the UNIPROT entry number.

The manuscript contains a number of formatting errors. Thus u is often used instead of the greak letter m (micro). See axis labels in Figure 3, 4 etc.. See also several instances in the text. Also very often a blank is missing between ‘the nominator and the unit’ (i.e. 10uM, line 239).  

Author Response

Review #3

Comments and Suggestions for Authors

In their manuscript entitled ‚Human cytomegalovirus essential tegument protein pp150 is amenable to targeting by sequence-specific peptides and ELP-conjugates’ the authors Mitra et al. investigated the potential of four pp150-derived peptides of between nine to eleven amino acids to inhibit HCMV replication in human foreskin fibroblasts. They observed that in case of one of these peptides, an inhibition with an IC50 of about 1 micromolar can be recorded. Furthermore, they observed that this peptide retains inhibitory activity when an ELP-tail is added to this peptide. However, this fusion reduces the inhibitory activity by a factor of 20-fold. All experiments appear sound and are properly described. The overall scope of the manuscript is however quite narrow.

Major points

  1. In the title, the authors write that pp150 is amendable to targeting by sequence-specific peptides, and in the discussion, ‘we screen the inhibitory efficiency of peptides that targeted the conserved region of pp150’. I consider that many readers (including myself) expect from the title that these peptides target pp150, namely that they somehow interact with pp150 and thereby block pp150 activity. Even when omitting the interaction expectation, targeting would still be read as pp150 being mechanistically targeted by these compounds. However, this is not the case at all here, since the manuscript investigates pp150-derived peptides, only. Whether these are able to block wild-type pp150 function is not at all investigated in the manuscript. I would therefore strongly suggest rewording the title and the manuscript with respect to this.

As suggested above, we have changed the manuscript title to “Tegument protein pp150 sequence-specific peptide block cytomegalovirus infection”.

  1. Figure 1 is very poor and doesn’t really explain/illustrate anything. I did look up the PDB entry 5VKU on which the illustration is based. I do concede that the structure of an entire virus is difficult to illustrate but if the authors aim to explain how pp150 interacts with the HCMV capsid and more precisely with the proteins MCP, SCP and the triplex proteins then they have to provide a better illustration. My suggestion would be to provide A) a sketch that roughly illustrates that pp150 sits on top of the triplexes and in between the MCP/SCP hexameric and pentameric complexes and B) a cartoon/ribbon illustration of pp150 with the areas/segments of pp150 of interest (CR1, CR2, Cys Tetrad 1 and Cys Tetrad 2) highlighted in color.

We have replaced Figure 1 with ribbon diagrams that were generated by PyMOL software to clearly illustrate that pp150 cluster above triplexes and extend towards nearby SCPs (Fig. 1A), and conserved regions in pp150nt (Fig. 1B). We also added a cartoon illustration to depict the triplex-pp150-SCP interactions (Fig. 1C).

  1. The labelling of the CR2 peptide with an elastin-like polypeptide (ELP) does not add any important additional information to the manuscript since the labelling is clearly detrimental to the inhibitory activity of the CR2 peptide (greater 20-fold reduction). The authors write that it is known that ELP fusions enhance in-vivo bioavailability. However, no experimental data are provided that investigate such an advantage in case of the CR2 peptide. In the competing interest section, the authors write that one coauthor is involved in a company that works on developing ELP drug delivery technology. Since ELP doesn’t provide any advantage in the context of the present study (on contrary), I would suggest that all experiments related to ELP are removed. Likewise, the authors could provide additional experiments that show a clear benefit for using ELP-CR2 fusion constructs.

We have removed ELP studies from this manuscript as they would be suitable for further investigations and future studies.

Minor points

  1. Table 1: The authors marked the sequences in parentheses with attributes such as ‘resembles CR1’ etc. Since some of these sequences are identical to the wild-type sequences, the word ‘resembles’ appears confusing. I would also suggest that the authors provide in the table the residues numbers of the sequence segments in relation to a defined UNIPROT entry.

We have removed the word ‘resembles’ and included HCMV pp150 UNIPROT entry (Q6SW99) residue numbers for the peptides in Table 1.

  1. Figure 2: The title of the figure reads: ‘Amino acid sequence and predicted secondary structure of pp150nt.’ The wording ‘predicted’ is confusing. In structural biology the word prediction relates to structural information that is not available from experimental data but which needs to be bioinformatically inferred. However, in case of pp150nt, several experimental structures are available, hence no prediction is needed. In would be better to use the wording: ‘... and structure-derived secondary structure...’. The authors also write that Figure 2 was generated with the PyMOL software. Is this correct? I am not aware that Pymol creates such illustrations. However, I might be mistaken here. In relation to sequence information, the authors should provide the UNIPROT entry number.

As suggested above we have changed the language of the figure legend to “Amino acid sequence and structure-derived secondary structure of pp150” (Fig. 2). The secondary structure is predicted by Yu et. al (PMID 28663444) and downloaded from Protein Data Bank (RSCB PDB ID: 5VKU). This figure is illustrated using BioRender. We have also added the UNIPROT entry number for HCMV pp150 tegument protein (UNIPROT ID: Q6SW99) for reference.

  1. The manuscript contains a number of formatting errors. Thus u is often used instead of the greek letter m (micro). See axis labels in Figure 3, 4 etc. See also several instances in the text. Also very often a blank is missing between ‘the nominator and the unit’ (i.e. 10uM, line 239).

We have corrected the spacings and (µ) symbols as suggested above.

Reviewer 4 Report

Human cytomegalovirus essential tegument protein pp150 is amenable to targeting by sequence specific peptides and ELP-conjugates

HCMV, a betaherpesvirus, establishes latent infections within most of the human population. Reactivation, as well as primary infections, can lead to severe diseases states, particularly in the immunocompromised. Currently, there is no universal cure or vaccine for herpesviruses and therefore, novel therapeutic strategies are of interest against these pathogens. Recent advancements in cryo-electron microscopy (cryo-EM) have provided atomic level views of almost all human herpesvirus capsids. Mitra et al., have capitalized on this structural information by generating peptides designed to block interactions between the capsid and pp150, an essential HCMV tegument protein. They identified one candidate, pep-CR2, which inhibited viral growth in HCMV-infected cells. They also developed an ELP-pep-CR2 conjugate with the goals of enhancing future in vivo delivery of the pep-CR2 drug candidate. Interestingly, the ELP itself also exhibited inhibitory activity. Overall, the manuscript is written well and adds to the growing number of studies describing peptides as potential antiviral therapeutic routes. The comments herein lie within some of the data interpretation, the methods description, and visualization/presentation of the data.

Major concerns

  1. How specific does pep-CR2 bind its target? Did the authors perform a sequence scramble control of pep-CR2 to assess specificity? These are standard negative controls used to ensure the observed inhibitory effect by the peptide of choice is not acting non-specifically, or by some other manner, such as through amino acid composition or secondary structure, and should be included.

  1. What are the biological phenotypes associated with either removal or perturbation of pp150 CR2 in infected cells? Are the results in Figure 5 consistent with perturbation/removal of the pp150 CR2 region in HCMV-infected cells? If so, these comparisons should be discussed in the text to help the reader draw conclusions regarding the pep-CR2 mechanism of inhibition.

  1. The authors report very different IC50’s between pep-CR2 (1.3 uM) and ELP-pep-CR2 (21 uM), indicating the addition of ELP reduces pep-CR2 potency. The authors discuss possible reasons for these difference in lines 386 – 395 and claim that despite these potential explanations, ELP-pep-CR2 will still likely result in enhanced in vivo properties (compared to pep-CR2; lines 397-399). While the ELP-conjugated route may have worked in other systems (ref 43), it is unclear what the outcomes will be in vivo for this system based on the data presented in this manuscript. Therefore, this claim should be toned down.

  1. The IC50 of the ELP-control is calculated from a sigmoidal curve fit that does not have data within the inflection point of the curve (Figure 6B). The authors should explain why this is the case or repeat the experiment to show datapoints throughout the entire inhibitory profile to report an accurate IC50. Otherwise, an IC50 should not be reported.

  1. Could it be the case that the observed reduction of viral growth by ELP-pep-CR2 in HCMV-infected cells is due to non-specific interactions since a much higher concentration of ELP-pep-CR2 was used (compared to pep-CR2) in these experiments? This would be in line with the fact that the ELP-control on its own was found to exhibit non-specific inhibitory effects on infected cells (line 406). If so, the authors should mention this possibility.

  1. There are a few experimental details that should be described altogether or in more detail to increase transparency.
    1. Preparation of peptides: the authors mention in the last line that the working concentrations of peptides had less than 0.1% DMSO, yet how was this done? The authors should include the buffer conditions used for the peptide studies.

  1. Microscopy: the authors should move the methods details regarding the GFP-pp150 experiment in Figure 5 from lines 255-258 in the results section to the methodology section to make these details easier to locate.

  1. Peptide inhibition studies: How were the inhibition experiments carried out in Figures 4 and 6? How was % inhibition calculated? Some of this experimental detail is given in the results section, yet this should be given its own section in the materials and methods. Additionally, it is not clear why an MOI of 3 was used for viability studies whereas an MOI of 0.1 was used for IC50 This detail should be added to the methods section.

  1. Generation of ELP-peptide construct: the authors should include the sequences of the primers used to clone this construct. It would also be helpful if the authors included the UniProt ID (or something similar) for the ELP protein construct used for this project, so one is easily able to locate the amino acid sequence of the entire protein used in this study.

  1. Statistics: the authors should provide the amount of biological and technical replicates performed for each of the analyses presented in the manuscript (either here or in the figure legends).

  1. Figure 1 is difficult to see, particularly Figure 1B. The authors should include a close-up view of the important pp150 regions so it is easier to see where those regions may be potentially interacting with other capsid proteins. The ray function can be used in PyMOL when exporting the figure to increase the resolution, which will also help the visualization. Such clarity in the visualization would help the reader draw conclusions based on the mechanism of pep-CR2 action.

Minor concerns

  1. It would be helpful if the authors placed the appropriate citations at the end of each corresponding sentence in the introduction paragraphs (lines 48-59 and 61-68) rather than cluster all of them together at the end of the paragraph.

  1. A more recent structure of the HCMV capsid which includes the structures of both pp150 and UL77/UL93/UL48 complex (HSV-1 capsid-associated tegument complex homologs) was recently published and should be added to the cryo-EM studies cited on line 79.

Reference: Li et al., Nat Commun. 2021 Jul 27;12(1):4538. doi: 10.1038/s41467-021-24820-3

  1. The legend in Figure 5 has two typos: one on line 282 (capital C is missing from the word cells) and it looks like it was added to line 284.

  1. Figure S1 mentions a scrambled peptide within the figure image, yet the manuscript does not discuss a scrambled peptide.

  1. Are the additional gel lanes in Figure S2 important/relevant for the paper? If not, they should be cropped out (ensuring it is made clear the gel was not tampered with to generate the figure). The figure legend says lanes 2 and 8-10 are NA, but it is still unclear why they are in the figure.

  1. The authors should double-check the amount of significant figures used to report peptide concentrations in figure legends 4 and 6.

Author Response

Review #4

Comments and Suggestions for Authors

Human cytomegalovirus essential tegument protein pp150 is amenable to targeting by sequence-specific peptides and ELP-conjugates

HCMV, a betaherpesvirus, establishes latent infections within most of the human population. Reactivation, as well as primary infections, can lead to severe diseases states, particularly in the immunocompromised. Currently, there is no universal cure or vaccine for herpesviruses and therefore, novel therapeutic strategies are of interest against these pathogens. Recent advancements in cryo-electron microscopy (cryo-EM) have provided atomic level views of almost all human herpesvirus capsids. Mitra et al., have capitalized on this structural information by generating peptides designed to block interactions between the capsid and pp150, an essential HCMV tegument protein. They identified one candidate, pep-CR2, which inhibited viral growth in HCMV-infected cells. They also developed an ELP-pep-CR2 conjugate with the goals of enhancing future in vivo delivery of the pep-CR2 drug candidate. Interestingly, the ELP itself also exhibited inhibitory activity. Overall, the manuscript is written well and adds to the growing number of studies describing peptides as potential antiviral therapeutic routes. The comments herein lie within some of the data interpretation, the methods description, and visualization/presentation of the data.

Major concerns 

  1. How specific does pep-CR2 bind its target? Did the authors perform a sequence scramble control of pep-CR2 to assess specificity? These are standard negative controls used to ensure the observed inhibitory effect by the peptide of choice is not acting non-specifically, or by some other manner, such as through amino acid composition or secondary structure, and should be included.

For the initial screening of peptides we used four different peptides; pep-CR1, pep-CR2, pep-Cysteine Tetrad1, and pep-Cysteine Tetrad2 (Table 1, Fig 3A). Out of these peptides, only pep-CR2 reduced virus titers, and the three other peptides did not affect virus growth. Thus pep-CR2 should be very specific to interfere with pp150-target binding.

We did not use scramble control to assess pep-CR2 specificity. We used FLAG sequence peptides as our control. FLAG sequences are very unique as they are not naturally present in human proteins. The FLAG sequence is highly charged because of aspartate and lysine residues. Control peptide-treated cells did not show any inhibition. Thus, if the peptide inhibitory effect was only dependent on amino acid composition or secondary structure, pep-control (FLAG sequence) treated cells should have shown some effect on virus titers.

  1. What are the biological phenotypes associated with either removal or perturbation of pp150 CR2 in infected cells? Are the results in Figure 5 consistent with perturbation/removal of the pp150 CR2 region in HCMV-infected cells? If so, these comparisons should be discussed in the text to help the reader draw conclusions regarding the pep-CR2 mechanism of inhibition.

We have added these details in the discussion. CR2 mutant phenotypes are well characterized. Earlier studies by Tandon and Mocarski have shown that two independent CR2 point mutants fail to support viral replication. These replication-defective CR2 mutants exhibit a phenotype that is the same as that of the virus that carried a complete deletion of UL32 ORF. The UL32 mutant virus-infected cells showed defects in the formation of vAC which was highly vesiculated and contained few nucleocapsids or complete virus particles in contrast to intact wild-type vAC (PMID 18653449).

The results in Figure 5 (now Fig. 6) are consistent in HCMV infected HFs with the perturbation of pp150 CR2 by peptides (pep-CR2) where the formation of vAC compromised and thereby affecting virus maturation.

  1. The authors report very different IC50’s between pep-CR2 (1.3 uM) and ELP-pep-CR2 (21 uM), indicating the addition of ELP reduces pep-CR2 potency. The authors discuss possible reasons for these differences in lines 386 – 395 and claim that despite these potential explanations, ELP-pep-CR2 will still likely result in enhanced in vivo properties (compared to pep-CR2; lines 397-399). While the ELP-conjugated route may have worked in other systems (ref 43), it is unclear what the outcomes will be in vivo for this system based on the data presented in this manuscript. Therefore, this claim should be toned down.

We have removed ELP studies from this manuscript as they would be suitable for further investigations and future studies.

  1. The IC50 of the ELP-control is calculated from a sigmoidal curve fit that does not have data within the inflection point of the curve (Figure 6B). The authors should explain why this is the case or repeat the experiment to show data points throughout the entire inhibitory profile to report an accurate IC50. Otherwise, an IC50 should not be reported.

 We have removed ELP studies from this manuscript as they would be suitable for further investigations and future studies.

  1. Could it be the case that the observed reduction of viral growth by ELP-pep-CR2 in HCMV-infected cells is due to non-specific interactions since a much higher concentration of ELP-pep-CR2 was used (compared to pep-CR2) in these experiments? This would be in line with the fact that the ELP-control on its own was found to exhibit non-specific inhibitory effects on infected cells (line 406). If so, the authors should mention this possibility.

 We have removed ELP studies from this manuscript as they would be suitable for further investigations and future studies.

  1. There are a few experimental details that should be described altogether or in more detail to increase transparency.

Preparation of peptides: the authors mention in the last line that the working concentrations of peptides had less than 0.1% DMSO, yet how was this done? The authors should include the buffer conditions used for the peptide studies.

We have added the description of peptide stock preparation and buffer conditions in materials and methods.

  1. Microscopy: the authors should move the methods details regarding the GFP-pp150 experiment in Figure 5 from lines 255-258 in the results section to the methodology section to make these details easier to locate.

 We have moved the method details in Fig 6 (previously Fig. 5) to the methodology section.

  1. Peptide inhibition studiesHow were the inhibition experiments carried out in Figures 4 and 6? How was % inhibition calculated? Some of this experimental detail is given in the results section, yet this should be given its own section in the materials and methods. Additionally, it is not clear why an MOI of 3 was used for viability studies whereas an MOI of 0.1 was used for IC. This detail should be added to the methods section.

Regarding Fig 6, we have removed ELP studies from this manuscript as they would be suitable for further investigations and future studies.

Regarding Fig 4, we have added experimental details in the materials and methods section. Briefly, we enumerated the number of foci for each concentration in the IC50 assay in triplicate wells and normalized the data (# of foci) in Graphad Prism taking the highest number of foci in the assay as 0% (164.66) inhibition and the lowest number of foci (2) as 100% inhibition, and plotted them on a bar graph. The calculation that prism uses is ((value - minOfValues) / (maxOfValues - minOfValues))*100.

Low MOI CMV infections are commonly used in an IC50 assay since it allows for multiple replication cycles in contrast to high MOI infections that only allow for one round of replication. For our initial screen (Fig 3) 10µM of pep-CR2 treatment at a high MOI of 3.0 showed 90% reduction in virus titer. To confirm that the treated concentration of pep-CR2 (10µM) is not cytotoxic and that the virus titer reduction is not influenced by the cytotoxic effects of pep-CR2, we performed the cell viability assay at a high MOI of 3.0.

  1. Generation of ELP-peptide construct: the authors should include the sequences of the primers used to clone this construct. It would also be helpful if the authors included the UniProt ID (or something similar) for the ELP protein construct used for this project, so one is easily able to locate the amino acid sequence of the entire protein used in this study.

We have removed ELP studies from this manuscript as they would be suitable for further investigations and future studies.

  1. Statistics: the authors should provide the amount of biological and technical replicates performed for each of the analyses presented in the manuscript (either here or in the figure legends).

We have added the number of replicates performed for each analysis in the methodology section. We used triplicates of each group in our experiments. For virus titers, we harvested the samples for each group in triplicates and used a duplicate of these triplicates for virus plaque assay.

  1. Figure 1 is difficult to see, particularly Figure 1B. The authors should include a close-up view of the important pp150 regions so it is easier to see where those regions may be potentially interacting with other capsid proteins. The ray function can be used in PyMOL when exporting the figure to increase the resolution, which will also help the visualization. Such clarity in the visualization would help the reader draw conclusions based on the mechanism of pep-CR2 action.

We have replaced Figure 1 with ribbon diagrams that were generated by PyMOL software to clearly illustrate that pp150 cluster above triplexes and extend towards nearby SCPs (Fig 1A), and conserved regions in pp150nt (Fig 1B). We also added a cartoon illustration to depict the triplex-pp150-SCP interactions (Fig 1C). We have applied the ray function in PyMOL to increase the resolution.

Minor concerns 

  1. It would be helpful if the authors placed the appropriate citations at the end of each corresponding sentence in the introduction paragraphs (lines 48-59 and 61-68) rather than cluster all of them together at the end of the paragraph.

 This has been addressed.

  1. A more recent structure of the HCMV capsid which includes the structures of both pp150 and UL77/UL93/UL48 complex (HSV-1 capsid-associated tegument complex homologs) was recently published and should be added to the cryo-EM studies cited on line 79.

 Reference: Li et al., Nat Commun. 2021 Jul 27;12(1):4538doi: 10.1038/s41467-021-24820-3

 We have added this citation in the introduction (fourth paragraph) where we discuss the cryo-EM structure of pp150.

  1. The legend in Figure 5 has two typos: one on line 282 (capital C is missing from the word cells) and it looks like it was added to line 284.

We have corrected the typographical errors in Fig. 6 (previously Fig. 5) legend in lines 312 and 314.

  1. Figure S1 mentions a scrambled peptide within the figure image, yet the manuscript does not discuss a scrambled peptide.

We have removed ELP studies and subsequent supplemental figures related to ELP from this manuscript as they would be suitable for further investigations and future studies.

Please refer to the discussion above in point #1 (Review 4- major concerns) where we explained the controls used for this study.

  1. Are the additional gel lanes in Figure S2 important/relevant for the paper? If not, they should be cropped out (ensuring it is made clear the gel was not tampered with to generate the figure). The figure legend says lanes 2 and 8-10 are NA, but it is still unclear why they are in the figure.

We have removed ELP studies from this manuscript as they would be suitable for further investigations and future studies. We have removed supplemental figures that were related to ELP studies.

  1. The authors should double-check the amount of significant figures used to report peptide concentrations in figure legends 4 and 6.

Regarding figure 6, we have removed ELP studies from this manuscript as they would be suitable for further investigations and future studies.

Round 2

Reviewer 2 Report

Revision has improved the overall quality and contents of the manuscript. 

Author Response

Thank you for your comments and suggestions that helped us improve the quality of the manuscript.

Reviewer 4 Report

The manuscript is very much improved and I appreciate the authors' responses and edits. I only have a few minor comments to improve clarity of the presentation of manuscript.

1) Figure 1 is still somewhat unclear. For those not familiar with looking at herpesviral capsid structures, it is not immediately evident that the blue cartoon shape in 1C representing pp150 is that of 3 pp150 proteins. 1A is 3 copies of pp150 (based on the HCMV cryoEM capsid structure), but that is not clear from the figure or the legend. Additionally, 1B shows one pp150 protein, rotated 180 degrees, to show both sides (I think this is what is shown). To make things consistent, I would make it clearer in the legend what the readers are looking at (for example, arrows to show if something is rotated or indicate on the figure pp150-1, 2 or 3 if there are multiple pp150 proteins). This is especially helpful when one is considering how the pp150 peptide binds to another capsid component to result in the observed phenotypes. 

2) In thinking about how pep-CR2 perturbs viral replication, it would be helpful if the authors speculated on this more in the manuscript text. The information provided by the authors in response to Reviewer 1, Comment 3 is very informative and helpful, but it did not make it into the Discussion section. By looking at just the capsid structure (without knowing that pp150 is actually a very large tegument protein and plays many roles during viral replication), one may expect the peptide blocks pp150 from capsid binding by preventing interactions with the triplexes, yet the location of CR2 in the structure appears quite distant from the triplexes (hard to completely tell without a surface model). Albeit, conformational changes may occur that could allow CR2 and the triplexes to come closer together, but that is not known. Alternatively, pep-CR2 could be binding to a capsid protein prior to their nuclear localization or capsid assembly that then prevent pp150 association. Furthermore, the peptide could be preventing other host or viral protein interactions with pp150 that would also result in a similar phenotype to what is observed in this study. Even if briefly discussed, including the details given in the author response to Reviewer 1, comment 3 would help improve the clarity of the findings, particularly given there are many options for how the peptide could be functioning. This information would help others put the data into context with the herpesviral replication process. 

3) Line 573: missing a Figure number - just says (Figure  ).

Author Response

Thank you for your comments.

1. We have modified Figure 1 to indicate where the trimer (1A, 1C) vs. the monomer (1B) version of the pp150 is depicted. We have also marked the 180-degree rotation in 1B. We have modified the legends accordingly.

2. We have expanded the discussion in the manuscript and included details that were earlier included in the response to Reviewer 1, Comment 3. We have also included some of the possibilities suggested by this reviewer as we found them very appropriate for discussion.

3) Line 573: missing a Figure number - just says (Figure  ).

This has been fixed.